# A Theoretical Conversation about Responses to Information Overload

**Amanda Lehman \*** and **Sophie Jo Miller**

University of Wyoming Libraries, Laramie, WY 82071, USA; smille63@uwyo.edu

**\*** Correspondence: amandarl@uwyo.edu

**Abstract:** In this study, information overload is viewed through the lenses of Library & Information Science and Communication Theory in order to offer recommended solutions for individuals experiencing overload. The purpose of this research was to apply LIS and COMM theories to the pathologies and symptoms of information overload as experienced by individuals in an increasingly digital world. Extant survey work was reviewed and updated with literature collected through limited keyword searches. The authors framed active responses to information overload through dimensions selected from the European Commission's Digital Competence Framework as applied to Al-Shboul & Abrizah's (2016) Modes of Information Seeking. Further study should focus on international perspectives and addressing disparities in access to information.

**Keywords:** information overload; communication theory; digital literacy; modes of information seeking

---

## 1. Introduction

Current research into information overload is diverse and multi-disciplinary. This paper begins by reviewing the current literature and scoping updated resources, then places digital competencies in a framework to suggest responses to information overload. The discussion updates the conversation between theoretical approaches to information overload in the areas of Library & Information Science [LIS] and Communication Theory [COMM]. Theoretical analysis of the concepts involved is based on recent reviews, surveys, and related literature discussion to justify the recommended solutions. Authors link diverse viewpoints in peer-reviewed research to broaden the understanding of information overload.

With a theoretical understanding reviewed, this paper recommends active responses to overload during information seeking, along with lines of further study. The European Commission's Digital Competence Framework (DigComp 2.0) [1] outlines basic skills to describe citizens' levels of information literacy (see Lucas & Moreira's discussion in chapter 7 of Marques & Batista) [2]. Information overload is often discussed as a symptom possibly related to a lack of digital or information literacy [3–5] and applying chosen competencies to modes of information seeking provides insight into active responses to overload. This model follows recommendations in the literature, acknowledges limitations, and provides solutions with a view to quick reference. The authors offer example actions to communicate digital literacy competencies (Information Data and Literacy, and Communication and Collaboration) [1] along with applicability for a range of information seeking modes.

## 2. Definitions

For the purposes of this paper, the definition of information overload comes from Bawden & Robinson (2009): "the point where information becomes a hindrance to the subject, even though the information is potentially useful. Information itself, therefore, must be defined at least in relation to

the subject who is seeking and/or receiving this data/input" [3]. The human information seekers in this discussion are referred to variously as subjects, patrons, users, clients, and individuals, the latter of which is the chosen term for this paper.

According to both Horrigan (2016) and Bawden & Robinson, "information overload" may not be the best term for what's going on. Horrigan [4] suggests "information burden", while Bawden & Robinson [3] suggest "information anxiety." The overall point is that information overload is not as simple as having "too much information." Rather, people can be overloaded by the quality and quantity of information itself, or by the pressures associated with information. Horrigan makes an important point that symptoms and consequences of information overload are exacerbated by having low or no access to information [4].

## 3. Method & Limitations

While framing this LIS/COMM-based review of information overload, the authors identified systematic reviews of research in the past 3–5 years of literature as redundant. Instead, a framed humanist review provides connections, surveying diverse sources for relevance, serendipity, and insight. Initial results follow the relevant discussion in sections of Marques & Batista and focus on literature reviews related broadly to the topic. Reviews are discussed to frame necessary concepts and recommended solutions. Connected resources range from Johannson's 2015 semantic inquiry [6] to Weinstein et al.'s 2016 study of adolescent behavior in the digital environment [7] to Pew Research Center reports on *Information Overload* [4] and *The Future of Well-Being in a Tech-Saturated World* [5]. The extant survey work done by information scholars emphasizes the need to build on empirical studies to form connections between theoretical discussions and real-world applications.

### 3.1. Data Collection

Review material was gathered by searching for identified keywords in LIS and COMM, then limiting results to the current literature with necessary references to authorities. Keyword searches were performed in the Primo Central portal, which indexes Scopus as well as numerous other collections (Information Providers and Collections are listed in ExLibris Group's Knowledge Center (https://knowledge.exlibrisgroup.com/Primo/Content_Corner/Central_Discovery_Index/ Documentation_and_Training/010CDI_-_The_Central_Discovery_Index/020CDI_Collection_Lists)). Because of the abundance of research on information overload, it was necessary to limit the articles being analyzed in depth. The first step was generating a list of keywords to focus on, then results were limited as discussed below. The list of chosen keywords is included in Table 1.

**Table 1.** Primary and related keywords.

| Primary Keywords | Related Keywords |
|---|---|
| Information Overload | Communication Overload |
| Communication Theory | |
| Technological Determinism | Marshall McLuhan |
| Cultivation Theory | George Gerbner |
| Shannon's Information Theory | |
| Universal design | Choice Overload |

### 3.2. Information & Overload

Identified keyword search combinations were chosen from initial survey and subject knowledge. Results were limited to LIS/COMM publications and filtered for peer review. Conditionally, if results numbered over 100, date ranges and further limits to the full text were added.

- SEARCH "information overload" and "decision making" (333 results, 64 limited)
- SEARCH "information overload" and Shannon (32 results)

- SEARCH "information overload" and "communication overload" (10 results)
- SEARCH "information overload" and universal (22 results)
- Select publications in the 2017–2020 date range were added back into the corpus for review despite immediately unavailable full-text.

*3.3. Information Overload and Communication Theory*

Following the methods listed in Section 3.1, keywords were searched in combination:

- SEARCH "information overload" and "cultivation theory" (76 results, 15 limited)
- SEARCH "information overload" and "technological determinism" (77 results, 28 limited)

*3.4. Post-Search*

Articles were selected according to thematic relevance and timeliness. The search results provided a corpus of information that fueled discussion and led to suggested active responses.

*3.5. Limitations*

This was not a PRISMA-based systematic review, a method identified as redundant with the thorough study and aggregation of this topic in recent work. Keyword searching, limiting to full text, and sorting do not assure a complete survey of results. The authors sought expertise through the literature and collaborative writing process; however, we acknowledge that our channels were intentionally limited, and the connections and solutions offered are not yet assessed.

The authors recognize a bias toward English language publications because the authors could not reliably review other languages for this work. Inherent in a study like this is a national bias towards familiar perspectives, which the authors attempted to mitigate by using international tools and standards including the EU Competence framework.

Access to information is a complex issue, and studies like this assume individuals have a basic level of access to information and communication technology. Without excluding socio-economic issues or denying major disparities supported by institutions in our country, the authors acknowledge that participation in the digital society is interdependent with privileges related to information access. A lack of access for a significant population mitigates the results in this paper and requires further inquiry.

## 4. Results

*4.1. Information in Context*

References to information overload are not a recent phenomenon. Historic examples of information overload are scattered throughout the literature. Katherine E. Ellison addresses this in her book, *Fatal News*, connecting modern media theorists to themes in Gulliver's Travels that culminate with the idea that "meaningful experience and true knowledge will somehow be drowned out by the meaningless and superficial noise caused by the proliferation of mass media" [8]. Ellison's book was published in 2006, and her focus was on events over 200 years earlier. Despite the time that has passed, tracing citations for Ellison's book connects these historic concepts to current studies in media scholarship, psychology, information overload, and further literature studies. The idea that people struggle with the abundance of information and with digging through the noise to find good information is still being discussed.

Understanding information overload as a centuries-old human concern informs the decision to tie it in with Marshall McLuhan's theoretical understanding of media and communication. Specifically, McLuhan suggested that advances in communication technology cause cultural change, with the Electronic Age pushing society into becoming a retribalized global village. The Electronic Age, starting in 1850, began with the invention of the telegraph, effectively ending the Print Age (sparked by the invention of the printing press in 1450 [9]. Ultimately, issues of cultural change and communication technology are more complicated than this, and scholars have gone back and forth trying to build

theoretical connections to McLuhan's approach since his death in the 1980s. For example, the 5th and 10th editions of *A first look at communication theory* connect McLuhan's view to Technological Determinism and Media Ecology, respectively, with the latter building on the former. However, his idea that human history can be divided into eras based on communication technology is not unsound and ties into the idea of information overload as an ongoing issue, especially as it has rapidly evolved in recent decades.

Terminology and concepts in the study of information overload are also rapidly evolving with study and use. Many terms for the concepts and challenges of information overload are present in the literature including:

- information glut
- information smog
- information burden
- information lack
- information overflow
- filter failure
- information anxiety
- overload anxiety
- choice overload
- digital stress

Terms focus on the negative effects of the state of digital saturation, but multiple results including the following from Bawden & Robinson point to a major factor in understanding overload, the digital divide: "The latest incarnation of the concept of the 'information poor'. This group has been identified in the library/information literature over many years, more often simply by assertion than from any evidence, and always in the context of their needing the services of library/information professions" Hayden & Bawden in [3]. While this discussion questions the tendency in LIS to focus on the digital divide, perhaps this discomfort will lessen considering COVID-19, social distancing, remote learning, health communication, and telehealth.

Information science is also home to a formative theory for these discussions: Shannon's model of communication and Theory of Information [10]. The theory is beautiful [11], and its study ranges broadly through the literature, offering a common basic understanding of mediums of communication, channels of information, and entropy as the noise in said channels which interferes with accurate communication. Relying on connections in the literature, we place information overload in context to explain solutions in the language of digital literacy. Though this approach is not novel, the relevance of grass-roots literacy initiatives [12] is growing during a time with many opportunities to improve digital literacy as a path to success in daily life (let alone during crisis events in 2020 as discussed by Kaufhold et al.) [13].

*4.2. Recent Summaries*

Because there is recent work aggregating overload resources by authorities in many professions, the review finds unexpected intersections in the metaphysical concept of information as a basis for evolution [6]. A tech blog post on smartphone usage discusses the adaptation of human consciousness to digital society [14], and Sthapit also finds evidence of successful information seeking due to digital adaptation [15].

The rest of Marques & Batista's aggregated study of this subject framed in the digital age [2] guides this review. Most chapters of this book address relevant subject matter and are best summarized in their front matter and overview chapter. The book offers "a triad of individual, organizational, and societal perspectives of this issue. Another triad is also used, namely the causes, the symptoms and solutions for this problem" [2]. Because this paper is not a book, a focus on the individual is selected, but organizational and societal perspectives are left for better-suited inquiry. Over-communication

and miscommunication may be related to a society of uncertainty which would increase anxiety and other symptoms of overload.

The discussion of the factors causing overload journeys through messaging and channel volume, information mismanagement, perpetual availability, and the digital divide. Berke, Akarsu, and Obay finish the discussion of factors with an economic study, stating that "[i]nformation overload is an important issue in the digital economy" [2], and highlighting consequences of disparities in information access and utilization. Their discussion, while oriented to organizational decision making, highlights the exponential possible growth of overload issues as the digital economy further influences the consumption and growth of information and communication technologies. Ganguin, Gemko, & Haubold, the final chapter, compare the decision-based German *Theorem of Competence* with the pragmatic Anglo-American *Model of Educational Media Literacies*. As stated, "[t]herefore, in this sense, it can be verified that media dependence is one problem related to the ability to decode information" [2]. This connects the semantic and mathematical modelling to factors that may be used to improve the common daily activities of many citizens [10,16,17].

Bawden & Robinson offer an in-depth summation of information contexts, pathologies, and "issues relating to the changing information environment with the advent of Web 2.0: loss of identity and authority . . . and the impermanence of information" [3]. The article offers solutions to two classes of problems stemming from the quantity and diversity of information, and from the changing information environment. While the first class is deeply discussed in later literature, the rapid changes continue to require unexpected approaches. The aggregated results of studies on smartphone use in 2017 offer one snapshot from the broader recent review. The article suggests that "our [smartphone] behaviors have mostly moved from compulsion to practical application". This idea offers a connection to many ongoing studies into digital well-being along the lines of Serrano-Puche's proposed digital diet in Marques & Batista [2].

Related summaries include Stephens et al.'s factors for reframing a model of communication overload through a subject-specific approach, focusing on the "multidirectional nature of communication (how people carry on multiple conversations with multiple others)" [18]. Johannsen also offers a framework and an in-depth review of literature related to Semantic Information in Nature (2015). The "model [takes] into account the common denominators for all information, which are Syntax, Semantics, and Pragmatics, as defined by semiotician and philosopher Morris" [6]. These definitions reach out to discussions of artificial intelligence modelling with Furlan, et al. [19], studies in decision science, linguistics, and more; however the syntax and semantics of information are highly relevant to discussions of the quality element to overload. Pragmatics also allows individuals to parse available information, connecting filtering and management solutions for overload issues. Though this framework was not implemented in the current discussion, it adds to the discussion.

*4.3. Recommendations in the Literature*

Literacy being a main goal of the publication, it is no surprise that solutions from Marques & Batista's compilation connect literacy to successful approaches to overload. Balance and simplicity are suggested to avoid information overload, as are team cognition, information management, a digital diet, along with various technologies and suggestions. The final chapter explores digital competencies from the German and Anglo-American approaches [2]. Ganguin, Gemko, and Haubold request an international exchange on media literacy and offer a pragmatic combination of approaches to media critique, self-organization, and self-determination which can "substitute the need of a 'digital-media-librarian' who arranges the information" [2]. This may appear counter to some points in the following discussion, but the values and framework discussed support consideration of recommended solutions to overload.

Bawden & Robinson (2009) review models of information behavior and suggestions from information managers, concluding with a need for further study including information history and diverse behaviors of information seeking. While much discussion aims to empower the individual, "[w]hen it comes to giving the answers, especially those that do not explicitly exist in the text corpus,

the advantages of a human expert are abilities such as explaining, combining complex answers, and abstract reasoning" [19]. These findings from Furlan et al. succinctly demonstrate that we cannot cut information purveyors out of the equation. So, assuming intervention in individual information processing, one can offer help guiding seekers from their point of need always towards digital competence.

Bawden & Robinson advise readers against the very goals identified and prioritized in this paper: "[A] quest to improve information literacy may simply be covering up a more fundamental need for improved literacy itself ... the way in which information is communicated, and knowledge resources accessed, may be an important part of this, but cannot be the whole" [3]. Solutions discussed follow themes of taking control of the flow of information, choosing resources and channels to follow intentionally, using web 2.0 tools, and a defacto method of satisficing. Considering active and emotional problem-solving methods discussed by Weinstein, methods with success usually include active responses. These solutions require a framework to propose useful responses based on competence building opportunities at the point of entry for most information-seeking individuals. The various modes of information seeking behavior bear discussion when considering solutions in conversation with self-motivated information seeking. As outlined in the best summative fashion found, Al-Shboul & Abrizah (2016) state: "Seven modes of active information seeking emerged from the research findings: (a) decision to seek information by the scholars or intermediary, (b) exploration, (c) monitoring, (d) accessing, (e) categorization, (f) purification, and (g) satisfaction" [20]. Sthapit reported a small but relevant case study of diverse tourists in 2019, saying "findings suggest that because of the growth and use of digital technology, today tourists are adaptive, continuously assessing [their plan], and are more receptive to the acquisition of new information" [15]. Stephens et al. commute the quantity and quality components of overload into an "availability–expectation–pressure pattern" [18] which connects back to individual perceptions and behavior.

*4.4. Competency-Based Responses*

These results demonstrate a common call for a deeper theoretical understanding of information overload [2,18]. The authors present a convergence of models proposed to elucidate productive responses to information overload. Specifically, the EU Commission framework was adapted to this conversation by focusing on the first two competence areas, *Information and Data Literacy*, and *Communication and Collaboration.* The authors consider the other three areas out of the scope of this inquiry; however, *Digital Content Creation* and *Safety* bear further study. Table 2 visualizes Al-Shboul & Abrizah's seven modes of information seeking with applicable competencies and active approaches to overload. Appendix A summarizes the competencies, which are applied to the tiered information scenarios. Active responses are suggested based on related competencies assigned to seven modes of information seeking.

**Table 2.** Modes of information seeking related to digital competencies.

| Modes of Information Seeking | Digital Competencies | Active Response to Overload |
| --- | --- | --- |
| Decision to seek information | N.A. | N.A. |
| Exploration | 1.1, 1.2 | Practice filtering, find experts |
| Monitoring | 1.1, 1.2, 1.3 | Assess new information to add to/replace existing information |
| Accessing | N.A. | N.A. |
| Categorization | 1.2, 1.3, 2.5 | Manage useful information |
| Purification (or reduction) | 1.1, 1.3, 2.6 | Strategic satisficing |
| Satisfaction | 1.1–1.3, 2.1–2.5 | (cumulative effect of competencies) |

Modes of information Seeking [20]-Digital Competencies & Responses [1].

The following discussion uses the literature to elaborate on the suggested active responses to information overload included in Table 2.

## 5. Discussion & Further Inquiry

The proposed responses below are recommended based on the assumption that the individual is requesting help in their information seeking practice. While these responses can be used by an individual working alone, those with lower levels of digital literacy may find it helpful to pursue active responses with the assistance of an information professional. In fact, as discussed by Furlan et al., the intervention of a human expert offers the information seeker the opportunity to discuss more complex aspects of digital literacy with an information professional, which ultimately allows for a higher level of comprehension. Active responses will develop digital competency, which in turn will decrease the individual's struggle with information overload.

### 5.1. Active Responses Related to Modes of Information Seeking

The individual decision to seek information is the necessary condition that must take place prior to engagement with information overload. Digital competencies related to the decision to seek information are ultimately the digital competencies associated with the other modes of information seeking. Responses cannot be recommended via digital competencies until the individual has begun actively seeking information.

Exploration of information is the base of competencies 1.1 *Browsing, searching and filtering data, information and digital content* and 1.2 *Evaluating data, information and digital content*. Following the decision to seek, individuals will articulate their information needs, create search strategies, evaluate results, and critically interpret results. Overload occurs when the amount of material returned is more than the individual can reasonably process in the time available. Therefore, individuals need ways to narrow their exploration to materials that are both relevant and comprehensible, whether that is by using specific search methodology or filtering technologies. Individuals who are still learning these competencies may also need to seek expert help.

Monitoring information is essentially knowledge upkeep, and it relies on the competencies used to seek information alongside 1.3 *Managing data, information and digital content*. In order to best follow new developments in information, it is necessary to manage the information that is already known. Encouragement to both add and strategically remove information from the overall body of knowledge as it becomes outdated or irrelevant will help keep information at a manageable level.

Accessing information requires a level of privilege, depending on what is being accessed; competency alone does not allow for access. For the purposes of this section, the focus is on full text content availability and access pathways. Full text access is directly related to an individual's ability to acquire good information. If an individual cannot read the full text, they cannot have a complete understanding of the information they are consuming. Especially in academic scholarship, full text content is often hidden behind a paywall. If an individual lacks the means to remove the paywall barrier (either the financial means or membership at an institution that pays for access), they are at a disadvantage when engaging in information seeking.

Access pathways are discussed by Horrigan [4], who separates them into "access abundance" (those with home broadband, tablet, and smartphone access) and "access scarcity" (those with none of the above); he notes that those who experience access scarcity reported higher levels of information overload. He discusses this with the understanding that overload is part of information anxiety (anxiety of not being able to access needed information) more than information overload. Access to information is necessary in order to move on to categorization and purification. Understanding that access is less about competency and more about having the physical and technological means to access information, the choice was made not to attach competencies to this section. This issue is also addressed in the limitations section.

Categorization of information is connected to competencies 1.2 *Evaluating data, information and digital content*, 1.3 *Managing data, information and digital content*, and 2.6 *Managing digital identity*. Part of

information overload can arise with platform fatigue [5]; individuals who choose to build a secure, reliable identity can use digital technology profiles to manage data and information channels [2]. Successful managers more easily establish frameworks for evaluating information in the first place.

Purification can be understood through simplification and reduction of information, supported by competencies 1.2 *Evaluating data, information and digital content, and* 2.1 *Interacting through digital technologies.* Where individuals may seek to purify their information channels by filtering or other blocking methods, relying on interactions and learning appropriate digital communication will support digital engagement and success. Purification can be related to the concept of *satisficing*, referring to the act of consciously making choices about what information to take in. Doing this strategically can help decrease information overload; doing it poorly can result in information avoidance [3].

Satisfaction is discussed in studies of tourism and decision making [15] and relies on the range of dimensions of digital competency. Initial approaches should thus build on early dimensions and can progress as individuals establish and own their personal information strategies. The range of responses to overload with other competencies will all increase satisfaction upon successful implementation.

## *5.2. Further Inquiry*

The simplest lines of further inquiry would expand on the full range of digital competencies, following modes of inquiry as discussed in Marques & Batista for the organizational and societal perspectives (also considering factors discussed in the Pew Report). Appropriately, the diversity of related subject matter defies quantification, and semantic connections offer a basic opportunity for expansion of the conversation between and across historic discipline lines. Individual solutions will depend on the individual, so understanding appraisals of technology [18], adaptive technologies, and Web 2.0 continue to be a highly relevant set of tools bearing consideration. Exploration of Web 2.0 society in relation to the full range of digital competencies (particularly digital content creation) is worth consideration as well, so as to better understand the connections between the consumption of information, the creation (collaborative or individual) of information, and information overload.

Once more it bears mention that the socio-economic impacts of access to digital society are more commonly recognized in academic and professional conversation. Horrigan, and Burke et al., both refer to pressing factors affecting diverse populations on the new landscape of digital society. The American Library Association recently published a statement emphasizing the call to address "the disparities in access to information" for marginalized populations, in this case, Black, Indigenous, and People of Color [21].

## 6. Conclusions

This exploration offers a novel consideration of a limited framework of solutions to overload in the digital age. This paper is an attempt to synthesize the information presently available in order to explore means of helping individuals decrease information overload. While the authors have approached this using modes of information seeking and EU standards for digital competency, these are not the only means by which to respond to information overload.

Information overload will continue to evolve over time, as it has done for the past few centuries. While individuals can take steps to decrease information overload, it may be difficult to avoid it completely, especially as technology continues to change and the amount of information available to us continues to grow. Issues of information overload also rely on solidly defining what we mean by overload. In the past, overload has been used as a catch-all for individual problems with levels of information and means of finding information. Moving forward, there is a need for widespread adoption of terms that differentiate between information overload (as well as communication and digital overload), information anxiety, information burden, and other terms that help categorize the contexts in which individuals struggle with information. There is no perfect solution to information overload; however, the suggested responses in this paper may help individuals prevent themselves from being completely overwhelmed by the information input and output which is increasingly expected on a daily basis.

**Funding:** This research received no external funding.

**Acknowledgments:** The authors appreciate the time and effort from their draft reviewers, Larry Schmidt and Michelle Green. They also extend thanks to University of Wyoming Libraries for providing access to the databases used for the keyword searches and literature review referenced in this paper.

**Conflicts of Interest:** The authors declare no conflict of interest.

## Appendix A

**Table A1.** DigComp 2.0-the Conceptual Reference Model–Dimensions 1–2.

| Competence Areas Dimension 1 | Competences Dimension 2 |
| --- | --- |
| 1. Information and data literacy | **1.1** Browsing, searching and filtering data, information and digital content<br>To articulate information needs, to search for data, information and content in digital environments, to access them and to navigate between them. To create and update personal search strategies.<br>**1.2** Evaluating data, information and digital content<br>To analyse, compare and critically evaluate the credibility and reliability of sources of data, information and digital content. To analyse, interpret and critically evaluate the data, information and digital content.<br>**1.3** Managing data, information and digital content<br>To organise, store and retrieve data, information and content in digital environments. To organise and process them in a structured environment. |
| 2. Communication and collaboration | **2.1** Interacting through digital technologies<br>To interact through a variety of digital technologies and to understand appropriate digital communication means for a given context.<br>**2.2** Sharing through digital technologies<br>To share data, information and digital content with others through appropriate digital technologies. To act as an intermediary, to know about referencing and attribution practices.<br>**2.3** Engaging in citizenship through digital technologies<br>To participate in society through the use of public and private digital services. To seek opportunities for self-empowerment and for participatory citizenship through appropriate digital technologies.<br>**2.4** Collaborating through digital technologies<br>To use digital tools and technologies for collaborative processes, and for co-construction and co-creation of resources and knowledge.<br>**2.5** Netiquette<br>To be aware of behavioural norms and know-how while using digital technologies and interacting in digital environments. To adapt communication strategies to the specific audience and to be aware of cultural and generational diversity in digital environments.<br>**2.6** Managing digital identity<br>To create and manage one or multiple digital identities, to be able to protect one's own reputation, to deal with the data that one produces through several digital tools, environments and services. |

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
