# Peer review of "A Theoretical Conversation about Responses to Information Overload"

_information, doi:10.3390/info11080379_

Round 1

Reviewer 1 Report

The manuscript presents the results of a review of the literature on information overload. The review is interesting and provides a relevant summary of the research in the area.

I just have two concerns:

My first concern relates to the organization of the paper, which I find slightly strange. Despite the article is a literature review, it also includes its own “literature review” (section 2). In its turn, Section 2 includes subsections on “recent summaries” (2.3) or “recommendations” (2.4). What is the difference between these “recent summaries” and those provided in the “results/discussion” of the paper? Actually, the “results” section is extremely short (15 lines). I would suggest the authors to consider whether to use section 2 only to define the concept of “information overload” and present a whole literature review as a “result” of their paper.

The abstract of the paper suggests that the purpose of the article is “to offer recommended solutions for individuals experiencing overload”. I think the paper falls short in reaching this goal. The Discussion (section 5) lists a set of “competencies” in information seeking. However, as a result of the review one would expect a list of recommendations for individuals about how to cope with information overload. I suggest the authors to develop further this part of the paper. This problem is partially linked to the previous one, since some “recommendations” are offered in the “literature review” (section 2).

Author Response

Point 1: My first concern relates to the organization of the paper, which I find slightly strange. Despite the article is a literature review, it also includes its own “literature review” (section 2). In its turn, Section 2 includes subsections on “recent summaries” (2.3) or “recommendations” (2.4). What is the difference between these “recent summaries” and those provided in the “results/discussion” of the paper? Actually, the “results” section is extremely short (15 lines). I would suggest the authors to consider whether to use section 2 only to define the concept of “information overload” and present a whole literature review as a “result” of their paper.

Response 1: We moved the majority of section 2 into the results, considering these comments.  This changed and simplified section titles and headings [Section 2 is renamed Definitions (line 41), Section 4 now begins at line 113 and includes the Information in Context (line 114), Recent Summaries (line 174), Recommendations in the Literature (line 223), and Competency-based Responses (line 265)].  This guidance applied to other comments and, we thought, lessened the need for major text revision.

Point 2: The abstract of the paper suggests that the purpose of the article is “to offer recommended solutions for individuals experiencing overload”. I think the paper falls short in reaching this goal. The Discussion (section 5) lists a set of “competencies” in information seeking. However, as a result of the review one would expect a list of recommendations for individuals about how to cope with information overload. I suggest the authors to develop further this part of the paper. This problem is partially linked to the previous one, since some “recommendations” are offered in the “literature review” (section 2).

Response 2: The introduction to section 5 beginning at line 282 adds justification and context to the offered recommendations.  Instead of editing for individual responses, we chose to depend on the intervention of an information professional as discussed in lines 283-289.  As stated, the re-organization of the sections also clarifies the recommendations. 

Reviewer 2 Report

In data collection the authors mention that the searches were performed in an institutional discovery search portal. It is important the authors to mention the resources that institutional discovery, because it seems very limited. Moreover, the authors should justify why they chose to collect data from institutional discovery tool and not from Scopus, Web of Science, Google Scholar or E-LIS. In addition, it would be useful to mention the link to institutional discovery search portal. Also, the authors results were limited to LIS/COMM journals and filtered for peer review why journal, but it is not clear if the results are limited to journal articles due to the limitations of the search or the authors decide to keep only the journal articles. It is important the authors to explain if these results are due to search limitation or if they decided to keep only the journal articles explain the reasons. The authors should also improve the reference section, for example in reference 2 it’s only the title of the book and not the author/s or the editor/s, in reference 4,5,6,7 they end the reference with semicolon (;) and not with full stop (.) as in other references. Also, in references 16, 18, the authors use capital letters instead of lower letters. In line 33 the authors mention see Lucas & Moreira’s discussion, since the authors refer to a book it is better to write the book pages in order the reader to locate that reference. In line 311 the authors referring is discussed widely in studies of tourism and decision making but they do not have reference in those studies.

The subject of the paper is very interesting and is a big issue those days, although the authors should present more details about the papers they used for the discussion, also the bibliographic references. In addition, the authors should expand their research with the help of Google Scholar or Scopus and draw conclusions that are international. Also, the authors in the results and discussion sections should provide more details about the articles they included and more specific results. The subject is very interesting, but it is important the authors to seriously revise the paper, draw international conclusions due to limitations the authors refer. In addition, it would be very useful the authors to read paper that are reviewing bibliography in order to improve their paper.

Author Response

Point 1:  In data collection the authors mention that the searches were performed in an institutional discovery search portal. It is important the authors to mention the resources that institutional discovery, because it seems very limited. Moreover, the authors should justify why they chose to collect data from institutional discovery tool and not from Scopus, Web of Science, Google Scholar or E-LIS.  In addition, it would be useful to mention the link to institutional discovery search portal.

Response 1: Since submission we were able to confirm that our portal indexes scopus.  We clarified the source of the institutional portal and provided a link to information on the index in Section 3.1, lines 71-72 and the accompanying footnote. 

Point 2: Also, the authors results were limited to LIS/COMM journals and filtered for peer review why journal, but it is not clear if the results are limited to journal articles due to the limitations of the search or the authors decide to keep only the journal articles. It is important the authors to explain if these results are due to search limitation or if they decided to keep only the journal articles explain the reasons.

Response 2: Line 217 changed “journals” to “publications” to indicate that other types of publications were included in reviewed search results.  The subject limitations scoped the results to a manageable size. 

Point 3: The authors should also improve the reference section, for example in reference 2 it’s only the title of the book and not the author/s or the editor/s, in reference 4,5,6,7 they end the reference with semicolon (;) and not with full stop (.) as in other references. Also, in references 16, 18, the authors use capital letters instead of lower letters. In line 33 the authors mention see Lucas & Moreira’s discussion, since the authors refer to a book it is better to write the book pages in order the reader to locate that reference. In line 311 the authors referring is discussed widely in studies of tourism and decision making but they do not have reference in those studies.

Response 3: These formatting changes were adopted as suggested, or further corrected, in the references section beginning at line 480.  The reference to tourism perspectives was clarified and a citation added in lines 328-329. 

Point 4: The subject of the paper is very interesting and is a big issue those days, although the authors should present more details about the papers they used for the discussion, also the bibliographic references. In addition, the authors should expand their research with the help of Google Scholar or Scopus and draw conclusions that are international.

Response 4: In response to point 1 we clarified the sources searched which included Scopus.  Google Scholar was not used for this paper because Scopus and other sources in the Primo Search Portal appeared to cover diverse resources. 

Point 5: Also, the authors in the results and discussion sections should provide more details about the articles they included and more specific results. The subject is very interesting, but it is important the authors to seriously revise the paper, draw international conclusions due to limitations the authors refer. In addition, it would be very useful the authors to read paper that are reviewing bibliography in order to improve their paper.

Response 5: In correcting the explanation of the research portal used, the authors confirmed that Scopus resources were included as a source of international perspectives.  Section 3.5 lines 97-100 was added to acknowledge a national bias. 

Round 2

Reviewer 2 Report

The authors fulfilled all the requested modifications.